# Early Return to Daily Life through Immediate Weight-Bearing after Lateral Malleolar Fracture Surgery

**DOI:** 10.3390/ijerph19106052

**Published:** 2022-05-16

**Authors:** Sang-June Lee, Youngrak Choi, Seongju Choi, Hoseong Lee

**Affiliations:** 1Department of Orthopaedic Surgery, Wonju Severance Christian Hospital, Yonsei University Wonju College of Medicine, Wonju 26426, Korea; lsjshock@gmail.com; 2Department of Orthopedic Surgery, Asan Medical Center, University of Ulsan College of Medicine, 88, Olympic-ro 43-gil, Songpa-gu, Seoul 05505, Korea; hosng@amc.seoul.kr; 3Department of Orthopaedic Surgery, Nowon Eulji Medical Center, Eulji University School of Medicine, Seoul 01830, Korea; seeds1617@gmail.com

**Keywords:** ankle fracture, lateral malleolus fracture, early rehabilitation, immediate weight-bearing

## Abstract

Lateral malleolus fracture is one of the most common fractures. However, there is controversy regarding the rehabilitation protocols used after surgery. In particular, the initiation point for weight-bearing has not been standardized. In the present study, we investigated the prognostic difference between immediate and delayed weight-bearing on lateral malleolus fractures. The medical records of matched patients in the immediate and delayed weight-bearing groups (50 and 50, respectively) were reviewed retrospectively. All patients were treated with open reduction and internal fixation using an anatomical locking compression plate with a lag screw. In the immediate weight-bearing group (IWB), tolerable weight-bearing (i.e., what can be endured immediately after surgery with crutches) was permitted. In the delayed weight-bearing group (DWB), weight-bearing was completely restricted for 4 weeks after surgery. Ankle motion exercise was permitted in both groups, starting from the day after surgery. Radiographic assessment data and clinical outcomes were reviewed between the two groups. No significant differences in radiographic assessments and complications were found between the two groups. Significant differences in terms of a shortened length of hospital stay and time to return to work with the IWB rehabilitation protocol compared with DWB were confirmed (6.0 vs. 9.2 days, *p* = 0.02 and 6.1 vs. 8.3 weeks, *p* = 0.02, respectively). A significant difference in sport factor was observed in the Foot and Ankle Outcome Score at 3 months postoperatively (72.3 vs. 67.4, *p* = 0.02). We found no significant differences between the two groups concerning postoperative radiological outcomes and complications. The benefits of shortening the time to return to work and length of hospital stay associated with the IWB rehabilitation protocol were confirmed. In conclusion, immediate weight-bearing is recommended in patients with lateral malleolus fracture after anatomical reduction and firm fixation by surgery.

## 1. Introduction

Ankle fracture is one of the most commonly reported fractures. Population-based epidemiological studies have shown an incidence of 168.7 in 100,000 fractures/year over a decade [1]. The most common type of fracture in all age groups was the lateral malleolar fracture, representing 55% of all ankle fractures [1]. During normal gait, the tibia plays a major role rather than the fibula. With the ankle joint in the neutral position, the weight distribution of the fibula reaches 6.4% [2]. Because of this anatomical characteristic, different postoperative care of lateral malleolar fractures was suggested compared with other types of ankle fracture [3].

A number of studies have recommended that patients begin weight-bearing soon after surgery for ankle fractures [4,5,6]. However, the initiation point for weight-bearing after ankle fracture during postoperative rehabilitation is still controversial [7,8,9]. A cross-sectional survey by expert orthopedic surgeons revealed 4.9 weeks for the average time of non-weight-bearing after fixation of ankle fractures in young, healthy patients and 7.6 weeks in older patients with medical comorbidities [9]. There is significant variation in the different non-weight-bearing periods of each surgeon. A standard rehabilitation protocol should be established because the period of non-weight-bearing leads to a lot of discomfort and loss of function for the patient. Many surgeons assume that early weight-bearing should be applied as soon as possible after surgery, but this application is limited owing to a lack of experience or concerns about displacement.

Therefore, the present study aimed to assess the clinical outcomes between immediate weight-bearing (IWB) and delayed weight-bearing (DWB) rehabilitation protocols in patients with a lateral malleolar fracture surgically treated with anatomical reduction and firm fixation. The hypothesis of the study was that IWB after fixation of a lateral malleolar fracture would not increase the rate of reduction loss in the postoperative radiographs as compared with that with DWB.

## 2. Materials and Methods

### 2.1. Study Population

The medical records of 312 patients who underwent open reduction and internal fixation (ORIF) of an unilateral ankle fracture were retrospectively investigated. The procedures were performed by a single surgeon from January 2015 to April 2019. Among the 312 patients, 223 had a lateral malleolus fracture treated with ORIF. All the patients who underwent surgery showed one or more of the following unstable findings before surgery as a surgical indication: displacement of >2 mm, fibular shortening, rotation, unbearable weight-bearing, and incongruent ankle mortise. Patients with an open ankle fracture, unavailable follow-up data for >12 months, low compliance with the rehabilitation protocol, combined ankle injury including complete rupture of the deltoid ligament, syndesmosis injury that underwent fixation, and/or osteochondral fracture of the talus were excluded from the analysis. Propensity score matching was performed between the two groups for potential confounders. The matched baseline variables included sex, age, and body mass index. Finally, 100 patients who had lateral malleolus fractures treated using an anatomical locking compression plate (LCP) were included (Figure 1). The matching ratio was 1:1 for the two groups based on the propensity score. Fifty who underwent immediate weight-bearing after surgery (IWB group) and 50 who did not undergo weight-bearing for 4 weeks after surgery (DWB group) were included and compared. In total, 100 patients were evaluated. A power analysis showed a power of 98.5% to detect a significant difference between the two groups (DWB versus IWB), assuming an alpha error level of 5% and a change in the opening gap width after surgery as the primary dependent variable. The mean ages of the patients in the IWB and DWB groups were 45.4 years and 48.4 years, respectively. Moreover, the mean follow-up duration was 28 months and 30 months, respectively. According to the Danis–Weber classification, the most frequent type of fracture was B (i.e., 46 and 45 fractures, respectively). No significant differences were found between the 2 groups regarding sex, age, body mass index, total number of screws, and fracture classification (Table 1).

### 2.2. Surgical Technique

All patients were treated with ORIF using an anatomical LCP (Arthrex, Naples, FL, USA) as a neutralization plate with a lag screw. After a longitudinal incision had been made along the midline of the fibular bone, the anatomical reduction was conducted and temporarily fixed using a Kirschner wire (1.6 mm). The anatomical reduction was fixed using an interfragmentary screw (3.5 mm cortical screw, Synthes, Solothurn, Switzerland). Subsequently, an anatomical LCP was applied to the appropriate location. We applied three/four distal screws and three proximal screws depending on the size and length of the fracture. The distal and proximal parts of all screws were 2.7 mm and 3.5 mm, respectively. The anatomical reduction was defined as an intraoperative fracture gap after fixation of ≤1 mm. Firm fixation was evaluated intraoperatively through observation of the fracture site with ankle joint motion.

### 2.3. Postoperative Management and Rehabilitation

After surgery, all the patients received a removable air cast walking brace (Seoul Prosthesis Corp., Yongsan, Seoul, Korea) for 4 weeks. In the IWB group, tolerable weight-bearing up to full body weight (i.e., what can be endured immediately after surgery with crutches) was permitted. Four weeks after surgery, in the outpatient department, all the patients of the IWB group completed full weight-bearing. In the DWB group, weight-bearing was completely restricted for 4 weeks after surgery. After 4 weeks postoperatively, tolerable weight-bearing was initiated and full weight-bearing was completed by 8 weeks. Ankle motion exercise was permitted in both groups on the day after surgery (Figure 2).

### 2.4. Outcome Measure

Ankle radiographs (i.e., in the anteroposterior, lateral, and mortise planes) were taken on the day after surgery, and at 2 weeks and 3, 6, and 12 months after surgery. The radiographic assessment, as a primary outcome, was focused on the time of bony union, loss of reduction, and formation of a new fracture. Loss of reduction was defined as a translation of >2 mm as compared with the postoperative immediate plain radiographs in 2 of 3 planes (anteroposterior, lateral, and mortise plane) at 3 and 12 months after the operation [3]. Moreover, the bony union was defined as the bridging of the trabeculae or osseous bone in 2 of the 3 planes (Figure 3) [10].

As secondary outcomes, the duration of hospitalization, time to return to work, and complications, especially surgical site pain, infection, wound dehiscence, and deep vein thrombosis were evaluated. Surgical site pain was defined as pain (i.e., >5 points on the visual analog scale) prolonged for >2 weeks after surgery. As a clinical score, the Foot and Ankle Outcome Score (FAOS) was evaluated at 3 and 12 months after surgery.

### 2.5. Statistical Analyses

For distribution assessment, the Kolmogorov–Smirnov test was used. The data are presented as means with 95% confidence intervals. The demographic and clinical variables of the two groups were compared using the chi-square test (categorical variables), and Student’s *t*-test (continuous variables). A *p*-value of <0.05 denoted statistical significance. Data were analyzed using the SPSS version 21.0 statistical software package (IBM Corp., Armonk, NY, USA).

## 3. Results

On the plain radiographs, the time of bony union did not show significant differences. The mean times to bony union were 4.2 and 4.3 months, respectively (*p* = 0.11). Significant differences in the length of hospital stay and time to return to work between the two groups were confirmed (IWB group vs. DWB group: 6.0 vs. 9.2 days and 6.1 vs. 8.3 weeks, respectively). Regarding complications, several patients reported surgical site pain (four and four cases, respectively; *p* = 0.36), surgical site infection (one and three cases, respectively; *p* = 0.34), and wound dehiscence (one and three cases, respectively; *p* = 0.35). Notably, loss of reduction, the formation of a new fracture, nonunion, and malunion were not reported in any of the patients. In the FAOS, a significant difference in sport factor was found at 3 months after the operation. The FAOS for sport were 72.3 and 67.4, respectively (*p* = 0.02; Table 2).

## 4. Discussion

Our principal findings are that there were no significant differences in postoperative radiological outcomes and complications between the two groups. Notably, although not statistically significant, fewer patients had complications (i.e., infection, wound problems) in the IWB group than in the DWB group. Furthermore, we confirmed the benefits of shortening the time to return to work and hospital stay with the IWB rehabilitation protocols. Our primary hypothesis that IWB after fixation of a lateral malleolar fracture would not increase the rate of reduction loss in the postoperative radiographs as compared with that with DWB was supported.

Previous studies have suggested that early postoperative weight-bearing after fixation of ankle fractures should be performed to prevent the development of iatrogenic joint stiffness, muscle atrophy, and deep vein thrombosis [4,11]. The randomized controlled study by Dehghan et al. revealed there was no difference in return to work, complication rate, infection, and loss of reduction between the early weight-bearing (weight-bearing and range of motion at 2 weeks) group and the late weight-bearing (non-weight-bearing and cast immobilization for 6 weeks) group [3]. Furthermore, better Olerud/Molander ankle function scores, SF-36, and ankle ROM were revealed in the early weight-bearing group [3]. Moreover, Simanski et al. suggested no disadvantages of early weight-bearing compared with delayed weight-bearing concerning the duration of hospitalization, pain intensity, time until return to work, and clinical scores [12]. According to a systematic review, no significant differences were found between the two approaches in terms of function, pain, range of motion, radiological assessment, complications, and return to work [13,14]. Another result of a meta-analysis of randomized controlled trial studies showed better results of early weight-bearing after ankle fracture surgery in terms of returning to work and daily activities compared with late weight-bearing [15]. A study of randomized controlled trials by the same author found that postoperative care systems such as unprotected weight loading and mobilization resulted in short-term improvements in functional outcomes, leading to an early return to work and sports, but the complications did not increase [16].

By contrast, other randomized controlled studies reported that postoperative early mobilization after fixation of ankle fractures should be restricted to protect against soft tissue damage and loss of reduction [17]. When applying early weight-bearing in patients, surgeons should exercise caution regarding the displacement of the fracture site and union. Such concerns may interfere with the decision to apply this rehabilitation approach, especially for less experienced surgeons. However, a cadaveric study demonstrated no significant fracture displacement, hardware failure, or occurrence of new fractures in cases of unstable ankle fracture after ORIF with an axial compression load [18].

The initiation of weight-bearing can be accelerated for numerous reasons. First, the application of early weight-bearing may accelerate the return of the patient to independent daily life and work. Gul et al. reported that patients who underwent early weight-bearing were associated with a significantly more rapid return to work than those who did not undergo early weight-bearing [11]. In the present study, significant differences in terms of a shortened hospitalization duration and time to return to work were observed. Second, joint stiffness may be prevented. Sondenaa et al. confirmed a limited ankle range of motion in a postoperative ankle fracture immobilized for 6 weeks with a plaster cast [19]. In our scoring system, we found a significant difference in the sport factor evaluated at 3 months after surgery. We assumed that the patient was unable to exercise because of the limitation of joint motion. Finally, other advantages (i.e., prevention of muscle wasting and osteoporosis) are expected. However, these effects have not been investigated thus far [3,15].

This study was characterized by limitations. First, we could not obtain sufficient data regarding the functional outcome. Data related to the range of joint motion and time of full weight-bearing would help confirm the benefits and verify the hypothesis. Therefore, parameters of muscle weakness and osteoporosis, muscle mass, and bone density after rehabilitation protocol must also be measured in future studies. Second, we could not assess patient compliance. The occurrence of pain may have affected the compliance of the patients. Therefore, investigators should devise strategies to improve patient compliance. Third, the study design was a retrospective chart review, which makes it impossible to randomly assign the patients to two groups. However, since the rehabilitation protocol was applied differently as IWB starting from 2016, there was no selection bias. There was not any indication of applying each rehabilitation protocol. Finally, our study only involved patients with lateral malleolar ankle fractures. Based on the data obtained from a randomized controlled trial, the postoperative rehabilitation method for each fracture pattern should be standardized.

## 5. Conclusions

We found no significant differences between the two groups in terms of postoperative radiological outcomes and complications. The benefits of shortening the time to return to work and hospital stay with the IWB rehabilitation protocol were confirmed. In conclusion, the application of immediate weight-bearing after surgery is recommended for patients with a lateral malleolus fracture who have undergone anatomical reduction and firm fixation.

## Figures and Tables

**Figure 1 ijerph-19-06052-f001:**
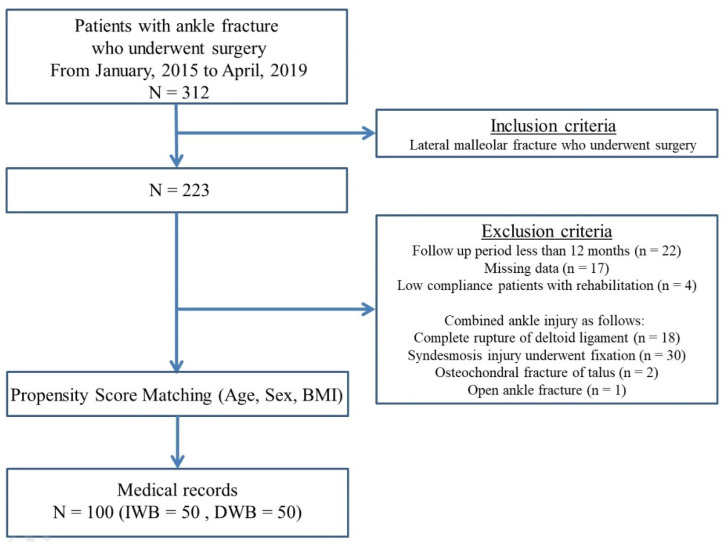
Flow chart of the study population.

**Figure 2 ijerph-19-06052-f002:**
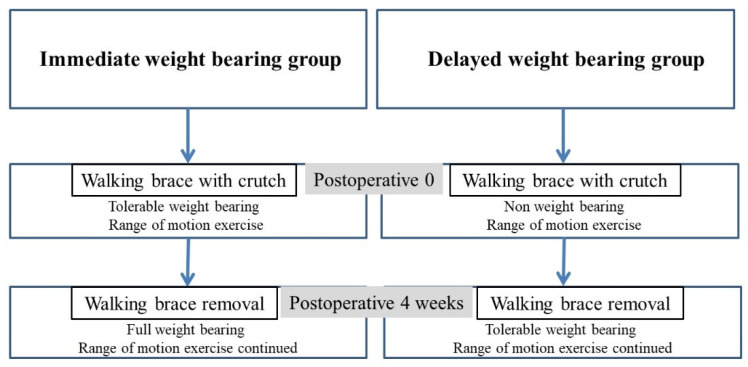
Postoperative rehabilitation protocol after fixation of the lateral malleolar fracture.

**Figure 3 ijerph-19-06052-f003:**
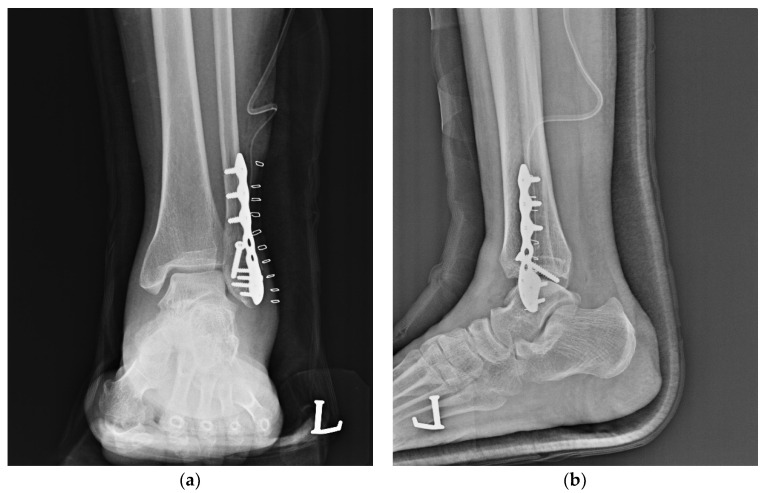
Postoperative ankle plain radiographs. (**a**) Anteroposterior view; (**b**) Lateral view.

**Table 1 ijerph-19-06052-t001:** Characteristics of the study population.

	IWB (95% CI)	DWB (95% CI)	*p*-Value
**Number of patients**	50	50	
**Sex (male/female)**	20/30	19/31	0.94
**Age (years)**	45.4 (30–64)	48.4 (26–59)	0.84
**Body mass index (kg/m^2^)**	27.7 (24–29)	27.5 (25–29)	0.88
**Weber-A/Weber-B/Weber-C**	4/46/-	5/45/-	0.99
**Lauge–Hansen classification** **(SA/SER/PA/PER)**	4/46/-/-	5/45/-/-	0.99
**Mean follow-up period (months)**	28.0 (20–32)	30.0 (21–36)	0.32
**Number of screws**	7.1 (7–10)	7.5 (7–9)	0.25

IWB, immediate weight-bearing; DWB, delayed weight-bearing; SA, supination adduction; SER, supination external rotation; PA; pronation adduction, PER; pronation external rotation.

**Table 2 ijerph-19-06052-t002:** Comparison of primary and secondary outcomes between the immediate and delayed weight-bearing groups.

	IWB (95% CI)	DWB (95% CI)	*p*-Value
** *Radiographic assessments* **			
**Time to bony union (months)**	4.2 (3–6)	4.3 (2.8–6.2)	0.11
**Loss of reduction (3 and 12 months)**	0/50 (0%)	0/50 (0%)	1.00
**Formation of a new fracture** **(3 and 12 months)**	0/50 (0%)	0/50 (0%)	1.00
** *Clinical outcome assessments* **			
**Hospitalization duration (days)**	6.0 (5–8.2)	9.2 (6–11.1)	0.02 *
**Time to return to work (weeks)**	6.1 (4.2–12.2)	8.3 (7–9.5)	0.02 *
**Surgical site pain**	4/50 (8%)	4/50 (8%)	0.36
**Surgical site infection**	1/50 (2%)	3/50 (6%)	0.34
**Wound dehiscence**	1/50(2%)	3/50 (6%)	0.35
**Deep vein thrombosis**	0/50 (0%)	0/50 (0%)	1.00
**FAOS for pain (3 months)**	64.5 (58.9–71.1)	68.5 (63.4–75.6)	0.16
**FAOS for symptoms (3 months)**	68.4 (63.2–72.4)	70.2 (63.8–75.3)	0.24
**FAOS for ADL (3 months)**	72.3 (69.3–76.2)	70.2 (68.3–75.2)	0.22
**FAOS for sport (3 months)**	72.3 (65.3–78.6)	67.4 (65.2–72.2)	0.02 *
**FAOS for QOL (3 month)**	75.2 (66.3–80.2)	74.4 (65.2–78.3)	0.31
**FAOS for pain (1 year)**	90.3 (80.2–93.2)	89.3 (83.3–92.1)	0.32
**FAOS for symptoms (1 year)**	87.3 (83.1–90.2)	88.3 (79.5–92.3)	0.24
**FAOS for ADL (1 year)**	90.5 (85.3–95.3)	89.5 (80.2–95.3)	0.25
**FAOS for sport (1 year)**	85.6 (80.2–91.3)	83.5 (82.3–92.3)	0.22
**FAOS for QOL (1 year)**	88.5 (83.2–94.3)	89.8 (80.2–91.5)	0.55

FAOS, Foot and Ankle Outcome Score; IWB, immediate weight-bearing; DWB, delayed weight-bearing; ADL, activities of daily living; QOL, quality of life, * *p* < 0.05.

## Data Availability

Not applicable.

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
