# Peer review of "Early Return to Daily Life through Immediate Weight-Bearing after Lateral Malleolar Fracture Surgery"

_ijerph, 2022, doi:10.3390/ijerph19106052_

Round 1
Reviewer 1 Report
Thank you very much for the submission of your precious and interesting work to the journal. I could reviewed the manuscript with great interest. The study is carried out with sound scientific methods. However, there are some point that should be cleared and revised to make the manuscript more readable.
- On the title, I am not sure if " propensity score matching analysis" should appear on the title. It is only a method to match experimental and control groups. I recommend to delete this part.
- The term "weight bearing" and "weight-bearing" are both used throughout the manuscript. it should be cleared and unify the term.
- Line#40-42. A reference for this incidence should be presented.
- Line#62-64. Null hypothesis and hypothesis should be different. On line#169-171, the authors presented that there hypothesis was that the clinical outcome of IWB rehabilitation protocol after fixation of lateral malleolar fracture is not inferior to the DWB protocol, which is the same with the null hypothesis on line#169-171. The authors should present their hypothesis on line#169-171 or change their null hypothesis. I recommend that the hypothesis should be more specific. As the authors presented that their primary outcome measure was the reduction loss on postoperative radiographs, I recommend the authors to change their hypothesis as the following. " The hypothesis of the study was that IWB after fixation of lateral malleolar fracture would not increase the rate of reduction loss on the postoperative radiographs as compared to that with DWB.
- Line#80 . The abbreviation should be spelled out the first time it appears in the text. The full term of LCP should be spelled out on line#80 not on line#88. Also the full term should be presented first and the abbreviation should be added inparentheses. For an instance, Locking Compression Plate (LCP). One line#88 LCP (Locking Distal Fibulaa Plate) is confusing. The abbreviation and the full term do not match here. Moreover, LCP is produced by Synthes not Arthrex. Figure 1 shows anatomical locking plate from arthrex. However, on line#92, cortical screw from Synthes is used as an interfragmentary screw. This should be cleared if this is true. If it is, the authors should explain if there was any reason to use screws with different manufacturer.
- Line#83-84. The authors explained that there was not any indication of applying each rehabiliation protocol. However, on lin#215-215, the authors explained that IWB started from 2016. It should be cleared if DWB was applied only during 2015 and IWB was started from 2015 through 2019. If this is true, it should be presented on line#83-84. Most of the times, when surgeon is sure of firm fixation, early weightbearing is allowed but when the firm fixation is doubted on osteoporotic bones, weightbearing is delayed which can cause selection bias for the study.
- Line#136-148. I recommend to move this part to the 2.1. Study population section on line#84.
- Line#151-153 should appear first at the result section as the radiological outcomes were the primary outcome measures.
- Line#158. "FAOSs" seems to stand for FAOS for sport. As this is quite confusing, please present FAOS for sport instead of FAOSs.
- Line#215-216. It is hard to say that there is no selection bias even for a randomized controlled study. I would like to recommend the authors to delete this part.
Reviewer 2 Report
Dear authors,
Your work is very interesting. I'm not an orthopedic specialist, but I'll give you a few comments about the formal part:
- TITLE: I suggest you to change the title, as in my opinion it doesn't capture enough attention to the topic you are talking about in the article.
- MATERIAL AND METHODS
- I believe that the design of your study is wrong, to do such a study and to avoid confounding factors as much as possible, you need to do a cohort study. With a retrospective study like yours, no randomized fracture treatment decisions were made and this may compromise the results.
- I believe the sample is too small to conduct a study that could bring significant results. The fact that the study does not have significant results is probably due to the low sample size. I recommend that you increase the number of people included in the study. I also recommend that you to conduct studies to assess the appropriate sample size for a study and include them in the methods.
- Regarding all tables, I suggest you to add a banding in clear-dark to allow an easier reading.
- I suggest you to review the final paper because there have been some problems, especially for the tables, in layout (some tables are halfway between one page and the next).
- DISCUSSION
- In literature there are numerous studies similar to yours, I suggest you search and add more citations because the bibliographic part is poor.
Reviewer 3 Report
Lee et al., retrospectively analyze recovery approaches in short and delayed weight-bearing following surgical stabilization of ankle fracture in their study population. Study analyses recommends immediate weight bearing.
Statistical approach is sound, selection of study population is sound. Although numerous other studies have reported similar findings, study supplementation provides appreciated data.
Round 2
Reviewer 2 Report
Dear authors,
I see you've absorbed the suggestions I've given you, but, as I told you previously, your work is very interesting, but I think it is not suitable for this journal because it is a purely clinical article. This is a Public Health journal, so I think you need to look for a journal that covers orthopedic topics.
